# Identification and Dissipation of Chlorpyrifos and Its Main Metabolite 3,5,6-TCP during Wheat Growth with UPLC-QTOF/MS

**DOI:** 10.3390/metabo12121162

**Published:** 2022-11-23

**Authors:** Lili Yu, Jia Li, Meiqin Feng, Qian Tang, Zejun Jiang, Hui Chen, Tingting Shan, Junhui Li

**Affiliations:** 1College of Animal Science and Food Engineering, Jinling Institute of Technology, Nanjing 210038, China; 2College of Life Sciences, China Jiliang University, Hangzhou 310018, China; 3Shandong (Linyi) Institute of Modern Agriculture, Zhejiang University, Linyi 276000, China

**Keywords:** wheat, chlorpyrifos, 3,5,6-TCP, dissipation, metabolite, UPLC-QTOF/MS

## Abstract

Ultrahigh-performance liquid chromatography system coupled to a hybrid quadrupole time-of-flight mass spectrometer (UPLC-QTOF/MS) technology was used to investigate the degradation and metabolism of chlorpyrifos during wheat growth by spraying plants with different doses of chlorpyrifos 7 days after the flowering and filling stage. We analyzed and identified chlorpyrifos metabolites in different parts of wheat in full-scan MS^E^ mode, and established a chlorpyrifos metabolite screening library using UNIFI software. The results show that the residues of chlorpyrifos in wheat ears, leaves, and stems exhibited a decreasing trend with the prolongation of application time, and the degradation kinetics could be fitted with the first-order kinetic equation C*_t_* = C_0_ e^−k^*^t^*. The initial residues of chlorpyrifos in different parts of the wheat were different, in the order of leaves > wheat ears > stems. The degradation rate of chlorpyrifos under field conditions is relatively fast, and the half-life value is 2.33–5.05 days. Chlorpyrifos can undergo a nucleophilic addition substitution reaction under the action of hydrolase to generate secondary metabolite 3,5,6-trichloro-2-pyridinol (3,5,6-TCP). The residual amount of 3,5,6-TCP in each part of wheat first showed an increasing trend and then decreased over time. It reached the maximum on the 3rd, 7th, or 11th day after application, and then gradually degraded. Considering that 3,5,6-TCP is a biomarker with potential threats to humans and animals, it is recommended that 3,5,6-TCP be included in the relevant regulations for dietary exposure risk assessment.

## 1. Introduction

Chlorpyrifos is an effective organophosphorus pesticide that is widely used worldwide [1,2]. Its degradation behavior in the environment and organisms, the types of metabolites, metabolic pathways, and the fate of metabolites in the environment affect the ecological environment and the safety of organisms [3,4]. There are many reports on the residual dynamics of chlorpyrifos in plants. In field trials, the rate of chlorpyrifos degradation in plants is fast, and the residual amount of chlorpyrifos is closely related to the number of applications, dosage, and weather conditions after application. The research of Shen Yan (2007) showed that spraying plants with the recommended doses of chlorpyrifos at different periods after flowering resulted in a downward trend of chlorpyrifos residues in wheat ears, decreasing in the order of 35 days after flowering > 14 days after flowering > 21 days after flowering > 7 days after flowering; the degradation half-life was relatively short, between 1.20 and 3.36 days. During the maturity period, chlorpyrifos residues were detected in different parts of wheat, mainly distributed in the bran and glumes, but not in flour [5]. Wu et al. (2012) found that the degradation trend of chlorpyrifos in green beans was in line with a first-order reaction kinetics equation, and the initial residual concentration of chlorpyrifos in green beans was low, ranging from 0.571 to 1.737 mg/kg. The degradation half-lives of chlorpyrifos in beans treated one- and twofold the recommended dosage were 1.6 and 1.5 days, respectively. During the safety interval (14 days), the residual amount of chlorpyrifos in green beans was lower than the safety limits of the European Union, South Korea, Japan, and other countries, further indicating that chlorpyrifos is an easily degradable pesticide [6].

Chlorpyrifos undergoes metabolic transformation in organisms, mainly through oxidation and hydrolysis, to generate corresponding metabolites [7,8,9]. The oxidative reaction of chlorpyrifos in the organism is mainly the metabolic reaction that first occurs under the catalysis of cellular P450s to form the unstable intermediate oxysulfide phosphate [10], which further undergoes oxidative desulfurization or dearylation to generate corresponding metabolites [11]. The oxidative desulfurization of chlorpyrifos is a toxicity-enhancing metabolic reaction that is mainly caused by the exchange of phosphorus atoms and oxygen atoms in the molecular structure to form the metabolite chlorpyrifos-oxon (CPO) [12]. A study by the United States Geological Survey found that the main oxidative metabolite of chlorpyrifos in the environment, CPO, is more toxic to animals than the parent pesticide is [13]. Antonious et al. (2017) reported that, under field conditions, after spraying chlorpyrifos on two kale varieties, the CPO metabolite was found. Although the residual concentration of CPO was lower than that of the parent pesticide chlorpyrifos, the degradation half-life (1.15–18.02 days) of CPO was longer than that of the parent pesticide chlorpyrifos (2.21–5.10 days), and its persistence and biological toxicity in plants were much higher than those of chlorpyrifos [14]. The dearylation of chlorpyrifos is a detoxification metabolic reaction through which chlorpyrifos can form metabolites diethylphosphate (DEP), diethylphosphorothioate (DETP) and 3,5,6-trichloro-2-pyridinol (3,5,6-TCP) [11]. In addition, chlorpyrifos can break the covalent bond between phosphorus atoms and leave groups under the catalytic action of various hydrolases, resulting in the less-toxic metabolites DEP, DETP, and 3,5,6-TCP [15,16,17].

QTOF/MS has the characteristics of high resolution, a wide mass range, high accuracy, and fast analytical speed, combined with the structural information obtained in MS/MS mode. It can obtain the accurate mass information of compounds through high sensitivity in the full-scan acquisition mode. In addition, QTOF/MS can optimize collision energy to obtain accurate mass precursor ions and fragment ions. Recently, QTOF/MS has been used as a powerful tool for multiple pesticide analysis and metabolite screening. Saito-Shida et al. (2016) developed a multiresidue method for the identification and quantification of 149 pesticides in four vegetables and fruits at a spiking level of 0.01 mg/kg using LC-QTOF/MS [18]. Similarly, Yang et al. (2018) developed a multiresidue detection method that used UPLC-QTOF/MS to identify and quantify 50 pesticides in fruits with high sensitivity and accuracy [19]. In another study, a multicomponent analytical method for the screening and quantification of pesticide residues in paprika samples via UPLC-QTOF/MS was developed and validated [20]. These results showed that QTOF/MS has high accuracy, sensitivity, and selectivity for the qualitative and quantitative aspects of compounds, and can be widely used in the analysis of pesticide residues, the identification of metabolites, and metabolomics research. However, there is still a lack of knowledge about the nature of the metabolite screening of chlorpyrifos during wheat growth.

The maximal residue limits (MRLs) of chlorpyrifos in wheat are 0.5 mg/kg in the Codex Alimentarius Commission (CAC), United States, Japan, and China [21,22,23,24]. Although MRLs are a feasible method to ensure the safe and effective implementation of pesticides, some metabolites in pesticides may be more toxic than the parent compound. However, few studies have focused on determining the MRLs of pesticide metabolites in food. It is, therefore, of great research significance to study the metabolic mechanism of chlorpyrifos in wheat.

The overall aim of the present study was to analyze and identify chlorpyrifos metabolites in different tissues of wheat through UPLC-QTOF/MS in full-scan MS^E^ mode, and to establish a chlorpyrifos metabolite screening library using UNIFI software. On the basis of accurate mass, retention time, and diagnostic ions, we found chlorpyrifos metabolite 3,5,6-TCP in different tissues of wheat in the open field. This research provides information to develop strategies for the safe use of chlorpyrifos under open-field conditions.

## 2. Materials and Methods

### 2.1. Reagents and Chemicals

A standard solution of chlorpyrifos (1000 mg/L) was purchased from the Agro-Environment Protection Institute, Ministry of Agriculture and Rural Affairs of China (Beijing, China). Commercial chlorpyrifos (45% emulsifiable concentrate (EC)) was obtained from Suzhou Jiahui Chemical Co., Ltd. (Suzhou, China). The 3,5,6-TCP standard (purity 99.6%) was supplied by Sigma-Aldrich (Shanghai) Trading Co., Ltd. (Shanghai, China). Organic solvents, including HPLC/MS-grade methanol, acetonitrile, formic acid, ethyl acetate, and ammonium acetate, were purchased from Thermo Fisher Scientific Corporation (Shanghai, China). Ultrapure water was collected from a Milli-Q system (Millipore, Billerica, MA, USA).

Standard stock solutions of chlorpyrifos and 3,5,6-TCP (100 mg/L) were prepared with LC-grade methanol and stored at −20 °C. These working standard solutions (0.001, 0.02, 0.05, 0.1, 0.2, 0.5, and 1 mg/L) were prepared by serially diluting the stock solution. Correspondingly, matrix-matched standard solutions of chlorpyrifos and 3,5,6-TCP (0.001, 0.02, 0.05, 0.1, 0.2, 0.5, and 1 mg/L) were prepared by serially diluting the working matrix standard solutions. These solutions were stored in the dark at 4 °C.

### 2.2. Field Experiment Design

Field trials were conducted from May to June in 2017 and 2018 at the Experimental Base of the Chinese Academy of Agricultural Sciences in Shunyi Farm, Beijing (116°33′ E, 40°13′ N). Weather data represent a monsoon climate of medium latitudes and were provided by the China Meteorological Administration. In 2017, the average daily temperature was 18.0–30.5 °C, the average daily sunshine hours were 14.38–14.93 h, and the average daily precipitation was 1.03–2.52 mm. In 2018, the average daily temperature was 18.5–30.5 °C, the average daily sunshine hours were 14.37–14.98 h, and the average daily precipitation was 1.09–2.65 mm. Chlorpyrifos was sprayed 7 days after the flowering and filling stage of wheat; plants were sprayed with clean water as a control. Treatment plots were sprayed with four different concentrations of chlorpyrifos. Plot 1 was used as the control. Plots 2–5 were sprayed with commercial 45% EC chlorpyrifos at the recommended dosage (450 mL/hectare), 2-fold the recommended dosage (900 mL/hectare), 5-fold the recommended dosage (2250 mL/hectare), and 10-fold the recommended dosage (4500 mL/hectare), respectively. Plots were randomly arranged, and each treatment was replicated three times. Wheat ear, leaf, and stem samples were taken on Days 0, 1, 3, 7, 11, 14, 21, 28, and 35 (harvest period) after application, and 50 plants were randomly sampled in each plot. These samples were placed in polyethylene bags and stored at −40 °C until analysis.

### 2.3. Sample Treatment

The homogenized sample (5 g) was placed in a 50 mL polypropylene centrifuge tube and extracted with 20 mL acetonitrile–water (50:50, *v*/*v*) for 30 min using an automatic shaker. Afterwards, a total of 4 g of MgSO_4_, 1 g of NaCl, 1 g of sodium citrate dihydrate, and 0.5 g of sodium hydrogen citrate sesquihydrate were added and shaken vigorously for 2 min, and then all the samples were centrifuged for 5 min at 6000 rpm. Then, 5 mL of the upper layer (acetonitrile) was transferred to a 15 mL centrifuge tube containing 150 mg of PSA, 150 mg of C_18_, and 900 mg of MgSO_4_. The tube was immediately vortexed for 1 min and centrifuged for 5 min at 6000 rpm. Then, 2 mL of supernatant-cleaned extract was evaporated to dryness at 60 °C. The dry extract was dissolved with 1 mL methanol, and the resulting solution was then filtered through a 0.22 μm nylon syringe filter (Jinteng, Tianjin, China) and analyzed with UPLC-QTOF/MS.

The extraction and clean-up of 3,5,6-TCP were performed as follows. The homogenized sample (5 g) was placed in a 50 mL polypropylene centrifuge tube and extracted with 20 mL of a mixture of acetonitrile and ethyl acetate (50:50, *v*/*v*), 10 mL of saturated sodium chloride solution, and 0.5 mL of 12 mol/L hydrochloric acid. The sample was vigorously shaken for 30 min. Subsequently, a centrifugation step (10,000 rpm, 5 min) was performed. Following this, 4 mL of the supernatant-cleaned extract was collected and transferred to an Oasis Prime HLB cartridge (Waters Corp, Milford, MA, USA) for purification. Then, the extract was evaporated to dryness at 45 °C and 200 mbar. The dry extract was reconstituted with 1 mL of methanol and the resulting solution was passed through a 0.22 μm nylon syringe filter (Jinteng, Tianjin, China) and analyzed using UPLC-QTOF/MS.

### 2.4. Instrumentation and Conditions

An ultrahigh-performance liquid chromatography system (ACQUITY UPLC I-CLASS, Waters Corp., Milford, MA, USA) coupled to a hybrid quadrupole time-of-flight mass spectrometer (VION IMS QTOF, Waters Corp., Milford, MA, USA) was used for this study. Sample separation was performed using a Waters ACQUITY UPLC HSS T3 analytical column (particle size 1.8 µm, 2.1 mm (i.d.) × 100 mm (length)). The mobile phase consisted of methanol (Solvent A) and 10 mmol ammonium acetate in water (Solvent B) applied at a flow rate of 0.45 mL/min in the following gradient mode: (i) 0 min (A–B, 2:98, *v*/*v*), (ii) 0.25 min (A–B, 2:98, *v*/*v*), (iii) 12.25 min (A–B, 99:1, *v*/*v*), (iv) 13 min (A–B, 99:1, *v*/*v*), (v) 13.01 min (A–B, 2:98, *v*/*v*) and (vi) 17 min (A–B, 2:98, *v*/*v*). The injection volume was set at 5 µL, and the column temperature was held at 45 °C.

The mass spectrometer in the positive electrospray ionization (ESI^+^) mode was used for the analysis of chlorpyrifos and its metabolites. The MS source conditions were as follows: capillary voltage, 1.0 kV; sample cone voltage, 40 V; nitrogen gas-flow of the nebulizer, 50 L/h; desolvation gas-flow, 1000 L/h; desolvation temperature, 550 °C; and source temperature, 120 °C. Spectral data was acquired in a mass range of *m*/*z* 50–1000 using full scan and the mass spectrometer elevated (MS^E^) experiment mode with an acquisition speed of 0.2 scan/s. In the MS^E^ function, the LE spectrum was recorded at 6.0 eV, and the HE spectrum was recorded with a collision energy ramped from 10 to 45 eV. Real-time calibration was performed with 200 pg/µL leucine–enkephalin (*m*/*z* 556.2766 in positive mode).

### 2.5. Method Validation

Recovery experiments were conducted to investigate the accuracy and precision of the method. Five replicates of spiked untreated wheat samples at five different levels (20, 50, 100, 200, and 500 µg/kg) were prepared with chlorpyrifos and 3,5,6-TCP working solutions in methanol. Then, the recoveries obtained with the spiked samples were compared with those of the matrix-matched calibration solutions. The limits of detection (LODs) for chlorpyrifos and 3,5,6-TCP were considered to be the concentration that produced a signal-to-noise (S/N) ratio of 3, and the limits of quantitation (LOQs) were assessed at an S/N ratio of 10. The linearity (R^2^) was evaluated using a matrix-matched calibration.

### 2.6. Statistical Analysis

The data were collected and processed using UNIFI™ 1.8.1 software (Waters Corp., Milford, MA, USA). Data were processed with a scientific library that was created inhouse containing a suspected database of chlorpyrifos metabolites (four library entities) with information about molecular structures, the exact mass analysis of precursor ions, fragment ions, adducts, and retention time in the database. In addition, in this study, the method conditions for pesticide screening to establish the scientific compound library were set according to Waters Corp. (Waters, Milford, MA, USA) [25]. The dissipation studies of residues of chlorpyrifos and its metabolite were performed using linear regression. Statistical analyses were performed using the PSAW Statistic 19.0 statistical software package (SPSS, Chicago, IL, USA). All data were statistically evaluated with one-way analysis of variance (ANOVA) followed by Duncan’s multiple-range test. The data are shown and were analyzed as micrograms per kilogram of matrix (mg/kg) on a dry-matter basis. All the values are reported as the means ± the standard deviation (SD) of five replicates.

## 3. Results and Discussion

### 3.1. Identification and Confirmation of Chlorpyrifos and Its Metabolites with UPLC-QTOF/MS

Table 1 shows the molecular formula, accurate mass, fragment ions, retention time and adducts of chlorpyrifos and its metabolite 3,5,6-TCP obtained in the full-scan MS^E^ mode of UPLC-QTOF/MS. The retention time of chlorpyrifos was 11.20 min; the characteristic fragment ions of chlorpyrifos with the same retention time obtained by QTOF/MS analysis were *m*/*z* 311.2577, *m*/*z* 197.9273 and *m*/*z* 96.9505. The retention time of the metabolite 3,5,6-TCP was 6.00 min, and the characteristic fragment ions of 3,5,6-TCP with the same retention time obtained by QTOF/MS analysis were *m*/*z* 184.0724, *m*/*z* 116.0693, and *m*/*z* 70.0646.

For all the possible metabolites of chlorpyrifos reported in Table 2, we established a metabolite screening library for all suspected metabolites of chlorpyrifos in different plant species to detect and identify the possible metabolites of chlorpyrifos during wheat growth. Then, we analyzed and identified chlorpyrifos metabolites in different parts of wheat through UPLC-QTOF/MS in full-scan MS^E^ mode, and established a pesticide metabolite screening library using UNIFI software. Table 2 shows the molecular formula and calculated exact mass of all possible metabolites of chlorpyrifos reported in the references. The identification and diagnostic proposal of metabolites were performed in accordance with the following criteria: (i) unique peaks in the processed sample compared with the blank sample; (ii) accurate mass deviation of precursor ion < 2 ppm mass error; (iii) retention time error of all samples < 0.1 min; (iv) at least ≥ 1 characteristic fragment ion by UNIFI^TM^ [26,27].

We analyzed and identified chlorpyrifos metabolite 3,5,6-TCP in different parts of wheat on the basis of accurate mass, retention time, adducts, diagnostic ions, and other standards. Since the reference substance of 3,5,6-TCP is commercially available, the further quantitative determination of 3,5,6-TCP in wheat samples was performed with UPLC-QTOF/MS. Figure 1A summarizes the UPLC-QTOF/MS-extracted ion chromatogram and MS^E^ spectrum of chlorpyrifos and its metabolite 3,5,6-TCP. Figure 1B shows the total ion chromatogram (TIC) and the low collision energy channel data for chlorpyrifos and 3,5,6-TCP. Chlorpyrifos presented an [M + H]^+^ peak as a base peak in the spectrum, and 3,5,6-TCP presented the sodium adduct [M + H]^+^ and [M + Na]^+^ as a base peak in which the hydrogen adduct might be the main form of chlorpyrifos and 3,5,6-TCP in mass spectrometry.

### 3.2. Method Validation for Chlorpyrifos and 3,5,6-TCP in Different Parts of Wheat

Five levels of chlorpyrifos and 3,5,6-TCP standard solutions were added to the blank wheat samples to verify the reliability of the method. The results are shown in Table 3. Within the range of 20–500 µg/kg, the concentrations of chlorpyrifos and 3,5,6-TCP exhibited excellent linearity with the peak area of the quantitative ion and the correlation coefficients (R^2^) ranged from 0.9958 to 0.9995. The mean recovery of chlorpyrifos and 3,5,6-TCP in different wheat matrices ranged from 63.38% to 102.13%, and the relative standard deviation (RSD) ranged from 1.76% to 8.99%. The LOD range of chlorpyrifos was 0.38–0.85 µg/kg, and the LOQ range was 1.30–2.70 µg/kg, which is lower than the MRLs established by the United States, the European Union, Australia, Japan, and China. The LOD range of 3,5,6-TCP was 2.90–4.50 µg/kg and the LOQ range was 11.00–14.00 µg/kg. The recovery rate and accuracy of this method are suitable for the determination of chlorpyrifos and 3,5,6-TCP content in different wheat matrices.

### 3.3. Dynamic Distribution of Chlorpyrifos in Different Parts of Wheat

The dynamic distribution of different concentrations of chlorpyrifos in different parts of wheat was fitted with the first-order reaction kinetics equation: C*_t_* = C_0_ e^−k*t*^ (where C*_t_* is the concentration of the pesticide at time *t*, C_0_ is the initial concentration of the pesticide, k is the degradation rate constant (1/d) and *t* is the number of days after application) [28,29]. *t*_1/2_ (degradation half-life) refers to the time required for the applied pesticide to be degraded by 50%. As presented in Table 4, in the field trials of 2017 and 2018, after spraying plants with different concentrations of chlorpyrifos pesticides, the degradation trends of chlorpyrifos in wheat ears, leaves, and stems all accorded with the first-order reaction kinetics equation. Here, the fitting degree was high, and the correlation coefficient ranged from 0.8646 to 0.9923. The field trial results (Figure 2, Figure 3 and Figure 4A) showed that, in 2017, the initial residues of chlorpyrifos in plants sprayed with 1-, 2-, 5- and 10-fold the recommended dosage were 2.649, 6.621, 18.316, and 46.653 mg/kg, respectively, in wheat ears (Figure 2(A-1)); 9.989, 12.656, 32.600 and 77.520 mg/kg, respectively, in the leaves (Figure 3(A-1)); and 1.515, 2.160, 7.033 and 19.333 mg/kg, respectively, in the stems (Figure 4(A-1)). In 2018, the initial residues of chlorpyrifos in plants sprayed with 1-, 2-, 5- and 10-fold the recommended dosage were 2.569, 5.860, 21.619 and 51.173 mg/kg, respectively, in wheat ears (Figure 2(A-2)); 10.288, 16.181, 34.887 and 81.440 mg/kg, respectively, in the leaves (Figure 3(A-2)); and 1.579, 2.701, 8.540 and 22.960 mg/kg, respectively, in the stems (Figure 4(A-2)). Field trial results in two localities for 2 years showed that the initial residues of chlorpyrifos in different parts of the wheat were different, and the distribution of initial residues for 2 years was in the order of leaves > wheat ears > stems. In addition, after application, the residual amount of chlorpyrifos in each part of the wheat plant showed a gradual degradation trend over time, dropping to a minimum at the maturity period (35 days after application). Our results are consistent with the report of Shen (2007), who indicated that, after spraying wheat plants with chlorpyrifos at different periods after flowering, the residual amount of chlorpyrifos in wheat ears, leaves, and stems showed a downward trend, with the initial deposition of chlorpyrifos in the different organs of wheat in the order of leaves > wheat ears > stems. The research results of Yang et al. (2018) showed that, after spraying rice plants with chlorpyrifos, the residual amount of chlorpyrifos in the various organs of the plants showed a single-peak curve and gradually degraded over time, and the initial deposits of chlorpyrifos were in the order of leaves > grains > stems > roots. Similarly, Zhang et al. (2012) reported that, after spraying rice plants with chlorpyrifos, the residual amount of chlorpyrifos in rice, husks, and straws dissipated gradually over time. The initial residues of chlorpyrifos in different tissues of rice were in the order of straw > husks > rice [30]. These results show that the degradation rule of chlorpyrifos in different parts of wheat is consistent and unaffected by changes in application dose and plant species.

Table 4 shows that the rate of chlorpyrifos degradation in each part of wheat was relatively fast, and the degradation half-lives of chlorpyrifos in wheat ears, leaves, and stems were 2.51–4.44, 3.15–5.05, and 2.21–3.46 days, respectively. The rate of chlorpyrifos degradation was fastest within 2–5 days after application, indicating that chlorpyrifos is an easily degradable pesticide, which may be related to its chemical properties, the characteristics of the applied plants, and environmental factors. In addition, in the early stage of application, the rate of chlorpyrifos degradation in each part of wheat was fast, and with the passage of time, it tended to be flat. The rule of chlorpyrifos degradation in various parts of wheat is not only affected by external environmental factors; the growth rate of the crop itself also greatly affects the persistence of pesticide residues in various parts of wheat. With the growth of plants, the pesticide is diluted by a larger surface area, and the pesticide content gradually decreases. However, as the plants grow, the cuticle becomes thicker in each part of the plant, especially the stems, as they contain more lignin, which means that pesticides are retained more strongly by mature plants than by young ones, and the pesticides that penetrate into the wheat tissue are less affected by external environmental factors; therefore, the degradation rate slows down. During the wheat harvest period (35 days after application), the residual amount of chlorpyrifos in wheat ears was below the national safety limit. Among treatments, in the group where chlorpyrifos was applied at the recommended dose, it was completely degraded, indicating that chlorpyrifos is an easily degradable and low-residue insecticide. This shows that spraying different doses of chlorpyrifos during the flowering and filling period of wheat is safe for wheat.

### 3.4. Metabolic Kinetics of 3,5,6-TCP in Different Parts of Wheat

In this study, UPLC-QTOF/MS and pesticide metabolite screening library analysis showed that, after chlorpyrifos was applied in the field, the 3,5,6-TCP metabolite could be detected in all parts of wheat. The dynamic relationship between 3,5,6-TCP residue and application time is shown in Figure 2, Figure 3 and Figure 4B. In 2017, the initial residues of 3,5,6-TCP in plants sprayed with 1-, 2-, 5-, and 10-fold the recommended dosage of chlorpyrifos were 0.057, 0.116, 0.657, and 1.604 mg/kg, respectively, in wheat ears (Figure 2(B-1)); 0.385, 0.889, 1.418 and 2.246 mg/kg, respectively, in the leaves (Figure 3(B-1)); and 0.000, 0.034, 0.193 and 0.604 mg/kg, respectively, in the stems (Figure 4(B-1)). In 2018, the initial residues of 3,5,6-TCP in plants sprayed with 1-, 2-, 5- and 10-fold the recommended dosage of chlorpyrifos were 0.048, 0.134, 0.807 and 1.909 mg/kg, respectively, in wheat ears (Figure 2(B-2)); 0.357, 1.336, 1.889 and 3.112 mg/kg, respectively, in the leaves (Figure 3(B-2)); and 0.000, 0.018, 0.262 and 0.740 mg/kg, respectively, in the stems (Figure 4(B-2)). The 2-year field data show that the initial residual concentrations of the 3,5,6-TCP metabolite in various parts of wheat were much lower than those of the parent pesticide chlorpyrifos, and the distribution of 3,5,6-TCP content in different parts of wheat in different dose groups was basically the same as that of the parent pesticide chlorpyrifos, i.e., in the order of leaves > wheat ears > stems. This may be because, under the same application conditions, the different structures of plant organs lead to differences in pesticide residues of plants. The critical surface tension, stratum corneum structure, and thickness of different plant organs are different and affect the adhesion and diffusion of pesticides on the plant surface [31,32]. As the main plant target organ for biotransformation and metabolism, leaves have a large surface area and a thin natural stratum corneum, which facilitate absorbing pesticides. Therefore, the initial residues of 3,5,6-TCP in leaves (0.357–3.112 mg/kg) and the persistence of pesticides in leaves were relatively high. During the wheat harvest period (35 days after application), the residues of 3,5,6-TCP in the leaves of plants sprayed with the recommended dosage were degraded, while the residues in the leaves of plants treated with 2–10-fold the recommended dosage were 0.038–0.588 mg/kg (Figure 3B). Compared with the leaves, stems have many similarities in the absorption of pesticides. However, the stem has a small surface area and thick cuticle, so its ability to absorb pesticides is relatively weak. The initial residual amount of 3,5,6-TCP in the stems was low (0.000–1.604 mg/kg), and the degradation rate was fast: 21 days after application, it was completely degraded in groups treated with 1- and 2-fold the recommended dose. During the harvest period, only a small amount of 3,5,6-TCP residues (0.012–0.033 mg/kg) could be detected in the groups treated with 5- and 10-fold the recommended dose (Figure 4B). As the edible part of wheat, wheat ears have a large surface area, but due to the enveloping glumes, the stratum corneum is thick, and the absorption capacity of both the parent pesticide and metabolites is relatively weak compared with that of the leaves. During the wheat harvest period, 3,5,6-TCP in wheat ears was completely degraded in the group sprayed with the recommended dose, and a small amount of 3,5,6-TCP was detected in the groups treated with 2–10-fold the recommended dose (0.031–0.093 mg/kg). The variation pattern of 3,5,6-TCP in each part of wheat was that it increased first and then decreased with degradation of the parent pesticide chlorpyrifos, with a slight fluctuation in the middle; it reached the maximal value on the 7th or 11th day after application, and then gradually degraded with the extension of time (Figure 2B).

Pesticides applied in the field can undergo metabolic transformation through oxidation, reduction, hydrolysis, photolysis, biodegradation, and other forms in plants through the action of enzymes or the influence of external environmental factors (temperature, precipitation, humidity, etc.), to produce specific metabolites [1,14,15,28,33]. In plant, soil, water and biological reaction systems, especially during photolysis, hydrolysis, and microbial metabolism, chlorpyrifos can be converted into more than 20 metabolites with a slow kinetic process and a half-life ranging from weeks to months [34]. Zhang et al. (2015) reported that, after spraying rice plants with chlorpyrifos, its 3,5,6-TCP metabolite was found in rice and rice stalks at a significantly lower concentration than that of the parent pesticide chlorpyrifos [35]. Antonious et al. (2017) reported that, after applying chlorpyrifos under field conditions, metabolites CPO and 3,5,6-TCP were found in two different kale varieties. Although the concentrations of the two metabolites were much lower than those of the parent pesticide chlorpyrifos, the degradation half-life of CPO (1.15–18.02 days) was much longer than that of chlorpyrifos (2.21–5.10 days), and that of 3,5,6-TCP (3.33–3.34 days) was similar to that of chlorpyrifos. The content of 3,5,6-TCP in kale also increased first and then decreased with the degradation of chlorpyrifos [14]. However, the results of Chai et al. (2009) showed that, after applying chlorpyrifos to green mustard and soil, no metabolites were found in green mustard, while 3,5,6-TCP was detected in soil. The degradation half-life of 3,5,6-TCP in soil was 3.6–9.4 days. The degradation rate was fast in the early stage and gradually slowed down after 21 days until the 77th day, at which point the 3,5,6-TCP in soil was basically degraded [36]. Similarly, in this study, we found that, after applying chlorpyrifos to wheat, chlorpyrifos would degrade under the action of natural conditions such as temperature, light, microorganisms, and rainwater, and produce the 3,5,6-TCP metabolite, which is highly toxic to the environment, water quality, animals, and the human body. Although the residual concentration of 3,5,6-TCP was lower than that of the parent pesticide chlorpyrifos, 3,5,6-TCP is a water-soluble compound with stronger polarity, greater toxicity, and greater mobility and durability [9,37,38]. Therefore, in order to reduce the risk of exposing chlorpyrifos metabolites to the ecological environment, and ensure the quality and safety of agricultural products, it is necessary to further clarify the metabolic pathways of chlorpyrifos in animals and plants.

Some studies have reported the metabolic pathway of chlorpyrifos in organisms. Racke (1993) found that chlorpyrifos was mainly metabolized into 3,5,6-TCP in the soil using isotope-labeled ^36^Cl-chlorpyrifos, and further verified that 3,5,6-TCP was mineralized and forms carbon oxides as secondary metabolites using ^14^C markers [34]. Lu et al. (2013) found that chlorpyrifos underwent hydrolysis under the action of strain DT-1 to produce 3,5,6-TCP and speculated that 3,5,6-TCP would then undergo three steps of dehydrogenation and produce secondary metabolites 2-hydroxypyridine, 5,6-dichloro-2-pyridinol and 6-chloro-2-pyridinol; lastly, the pyridine ring may be broken and degraded into small molecular compounds [39]. These research results are consistent with our predictions. After spraying wheat with chlorpyrifos, chlorpyrifos undergoes a nucleophilic addition substitution reaction under the action of organophosphorus hydrolase to generate the metabolite 3,5,6-TCP (Figure 5) [14,35]. With a continuous increase in OH concentration in the medium, the rate of hydrolysis accelerates, and the concentration of 3,5,6-TCP continues to increase. As time goes on, 3,5,6-TCP is affected by multiple factors such as its own properties, natural conditions, and crop characteristics, and may be further mineralized to produce small molecular compounds such as water, CO_2_, and ammonium ions [40].

## 4. Conclusions

In conclusion, our study is the first to reveal the metabolic mechanism of chlorpyrifos during wheat growth in addition to the dissipation kinetics in wheat of the parent chlorpyrifos and its metabolite 3,5,6-TCP. The degradation dynamics of chlorpyrifos could be fitted by the first-order kinetic equation C*_t_* = C_0_ e^−k^*^t^*. The initial residues of chlorpyrifos in different organs of wheat were different at the early stage of application, which was in the order of leaves > wheat ears > stems. Chlorpyrifos was hydrolyzed to 3,5,6-TCP which was further dissipated. 3,5,6-TCP first showed an increasing trend and then a decreasing trend over time. In addition, it reached the maximum on the 3rd, 7th, or 11th day after application, and then gradually dissipated.

The primary metabolic pathway of chlorpyrifos was initially verified, but studies on the secondary product’s metabolic pathway have not been further verified. Further research is still needed to reveal the metabolic pathway of chlorpyrifos in plants. Considering that 3,5,6-TCP is a biomarker with potential threats to humans and animals, the establishment of MRLs for 3,5,6-TCP should be considered during dietary exposure risk assessment.

## Figures and Tables

**Figure 1 metabolites-12-01162-f001:**
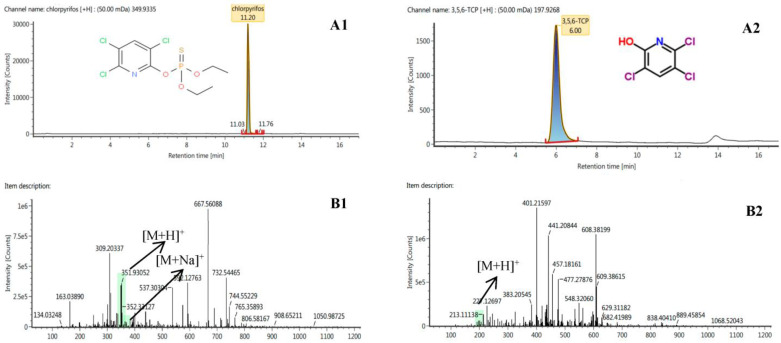
UPLC-QTOF/MS-extracted ion chromatogram and MS^E^ spectra of chlorpyrifos and 3,5,6-TCP: (**A1**) chlorpyrifos standard at 100 ug/kg in wheat sample; (**A2**) 3,5,6-TCP standard at 100 ug/kg in wheat sample; (**B1**) the low collision energy adducts of H^+^, Na^+^ chlorpyrifos; (**B2**) low-collision-energy adducts of H^+^, 3,5,6-TCP.

**Figure 2 metabolites-12-01162-f002:**
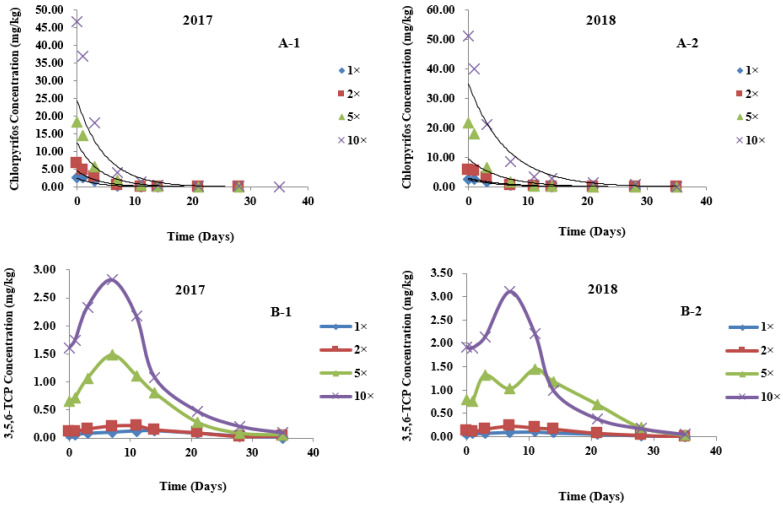
Degradation curves of chlorpyrifos and its metabolite 3,5,6-TCP at different applied dosages in wheat ears. (**A-1**) Degradation curves of Chlorpyrifos at different applied dosages in wheat ears in 2017; (**A-2**) Degradation curves of Chlorpyrifos at different applied dosages in wheat ears in 2018; (**B-1**) Degradation curves of 3,5,6-TCP at different applied dosages in wheat ears in 2017; (**B-2**) Degradation curves of 3,5,6-TCP at different applied dosages in wheat ears in 2018. 1×: recommended dosage. 2×: twofold recommended dosage. 5×: fivefold recommended dosage. 10×: tenfold recommended dosage.

**Figure 3 metabolites-12-01162-f003:**
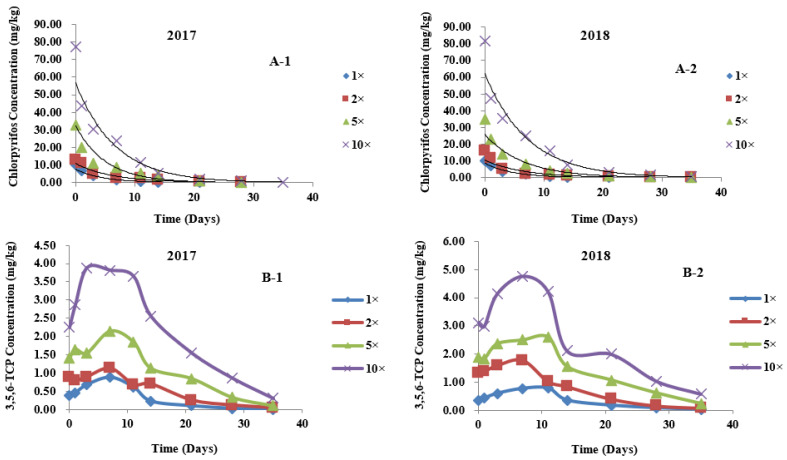
Degradation curves of chlorpyrifos and its metabolite 3,5,6-TCP at different applied dosages in leaves. (**A-1**) Degradation curves of Chlorpyrifos at different applied dosages in leaves in 2017; (**A-2**) Degradation curves of Chlorpyrifos at different applied dosages in leaves in 2018; (**B-1**) Degradation curves of 3,5,6-TCP at different applied dosages in leaves in 2017; (**B-2**) Degradation curves of 3,5,6-TCP at different applied dosages in leaves in 2018. 1×: recommended dosage. 2×: twofold recommended dosage. 5×: fivefold recommended dosage. 10×: tenfold recommended dosage.

**Figure 4 metabolites-12-01162-f004:**
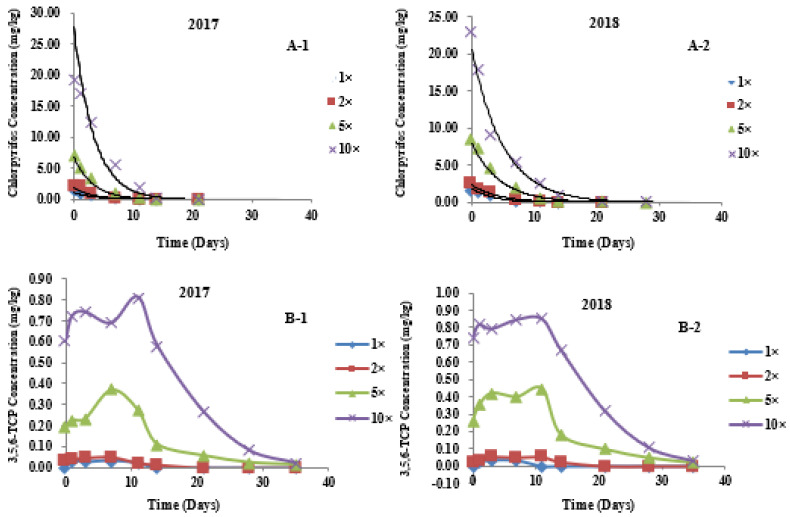
Degradation curves of chlorpyrifos and its metabolite 3,5,6-TCP at different applied dosages in stems. (**A-1**) Degradation curves of Chlorpyrifos at different applied dosages in stems in 2017; (**A-2**) Degradation curves of Chlorpyrifos at different applied dosages in stems in 2018; (**B-1**) Degradation curves of 3,5,6-TCP at different applied dosages in stems in 2017; (**B-2**) Degradation curves of 3,5,6-TCP at different applied dosages in stems in 2018. 1×: recommended dosage. 2×: twofold recommended dosage. 5×: fivefold recommended dosage. 10×: tenfold recommended dosage.

**Figure 5 metabolites-12-01162-f005:**
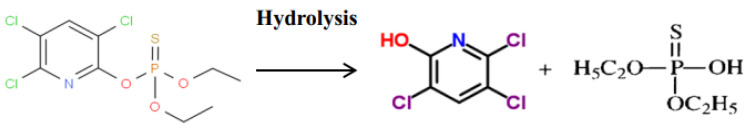
Hydrolyzed metabolic pathway of chlorpyrifos in wheat samples.

**Table 1 metabolites-12-01162-t001:** Accurate UPLC-QTOF/MS mass measurements of chlorpyrifos and 3,5,6-TCP in wheat samples.

Compound	Retention Time (min)	Formula	Observed Neutral Mass (Da)	Fragments	Adducts
Chlorpyrifos 3,5,6-TCP	11.20 6.00	C_9_H_11_Cl_3_NO_3_PS C_5_H_2_ Cl_3_NO	349.9335 197.9268	311.2577 197.9273 96.9505 184.0724 116.0693 70.0646	+H, +Na +H

**Table 2 metabolites-12-01162-t002:** List of reported metabolites of chlorpyrifos in references, their formula, and calculated exact masses.

Compound	Formula	Exact Mass (Da)
3,5,6-TCP	C_5_H_2_Cl_3_NO	198.4345
DEP	C_4_H_11_O_4_P	154.1015
DETP	C_4_H_11_O_3_PS	170.1671
CPO	C_9_H_11_Cl_3_NO_4_P	332.9491

**Table 3 metabolites-12-01162-t003:** R^2^, recoveries, LOD and LOQ of chlorpyrifos and its metabolite 3, 5, 6-TCP in different matrices (*n* = 5). Values (mean ± SD) in the same row.

Compound	Matrix	R^2^	Average Recovery and Standard Deviations (%)					LOD (µg/kg)	LOQ (µg/kg)
			Spiking Level (µg/kg)						
			20	50	100	200	500		
Chlorpyrifos	Wheatear	0.9995	72.00 ± 2.18	71.33 ± 6.43	72.67 ± 7.57	82.83 ± 1.76	102.13 ± 3.35	0.38	1.30
	Leaf	0.9974	76.67 ± 5.77	83.33 ± 5.03	85.00 ± 2.00	85.50 ± 2.78	76.73 ± 4.27	0.85	2.70
	Stem	0.9983	77.00 ± 3.29	78.46 ± 5.87	79.16 ± 7.22	85.45 ± 2.77	89.53 ± 4.88	0.70	2.20
3, 5, 6-TCP	Wheatear	0.9977	66.33 ± 6.24	70.01 ± 7.35	72.37 ± 8.35	83.57 ± 7.37	83.36 ± 8.99	2.90	11.00
	Leaf	0.9958	65.79 ± 5.34	63.38 ± 4.26	69.24 ± 6.38	72.43 ± 8.46	77.84 ± 6.59	4.50	14.00
	Stem	0.9983	67.86 ± 5.76	71.47 ± 7.58	73.15 ± 5.37	81.38 ± 3.58	87.67 ± 5.68	4.25	13.50

LOD: The limit of detection; LOQ: The limit of quantitation; 3,5,6-TCP: 3,5,6-trichloro-2-pyridinol; SD: Standard Deviation.

**Table 4 metabolites-12-01162-t004:** Degradation kinetics of chlorpyrifos at different applied dosages in different parts of wheat.

Matrix	Time	Treatment	First-Order Kinetic Equation	C_0_ (mg/kg)	R^2^	K (1/d)	t_1/2_ (d)
Wheatear	2017	1×	C*t* = 2.4668 e^−0.2765*t*^	2.4668	0.9201	0.2765	2.51
		2×	C*t* = 4.5245 e^−0.2609*t*^	4.5245	0.9638	0.2609	2.66
		5×	C*t* = 12.3650 e^−0.2510*t*^	12.3650	0.9653	0.2510	2.76
		10×	C*t* = 24.3582 e^−0.2080*t*^	24.3582	0.9265	0.2080	3.33
	2018	1×	C*t* = 2.4479 e^−0.1921*t*^	2.4479	0.9693	0.1921	3.61
		2×	C*t* = 2.9560 e^−0.1653*t*^	2.9560	0.8646	0.1653	4.19
		5×	C*t* = 9.3164 e^−0.1775*t*^	9.3164	0.8839	0.1775	3.90
		10×	C*t* = 34.7860 e^−0.1562*t*^	34.7860	0.9595	0.1562	4.44
Leaf	2017	1×	C*t* = 7.6341 e^−0.2199*t*^	7.6341	0.9771	0.2199	3.15
		2×	C*t* = 10.8377 e^−0.1580*t*^	10.8377	0.9726	0.1580	4.39
		5×	C*t* = 32.4195 e^−0.1954*t*^	32.4195	0.9381	0.1954	3.55
		10×	C*t* = 56.8813 e^−0.1494*t*^	56.8813	0.9898	0.1494	4.64
	2018	1×	C*t* = 8.6738 e^−0.2117*t*^	8.6738	0.9863	0.2117	3.27
		2×	C*t* = 10.5375 e^−0.1516*t*^	10.5375	0.9778	0.1516	4.57
		5×	C*t* = 25.7491 e^−0.1512*t*^	25.7491	0.9909	0.1512	4.58
		10×	C*t* = 62.3189 e^−0.1372*t*^	62.3189	0.9919	0.1372	5.05
Stem	2017	1×	C*t* = 1.0919 e^−0.2687*t*^	1.0919	0.8979	0.2687	2.58
		2×	C*t* = 1.8643 e^−0.2954*t*^	1.8643	0.9253	0.2954	2.35
		5×	C*t* = 6.8317 e^−0.3142*t*^	6.8317	0.9685	0.3142	2.21
		10×	C*t* = 27.7730 e^−0.2979*t*^	27.7730	0.9742	0.2979	2.33
	2018	1×	C*t* = 1.8867 e^−0.2950*t*^	1.8867	0.9893	0.2950	2.35
		2×	C*t* = 2.3523 e^−0.2311*t*^	2.3523	0.9799	0.2311	3.00
		5×	C*t* = 8.0059 e^−0.2213*t*^	8.0059	0.9843	0.2213	3.13
		10×	C*t* = 20.6520 e^−0.2002*t*^	20.6520	0.9923	0.2002	3.46

1×: recommended dosage. 2×: twofold recommended dosage. 5×: fivefold recommended dosage. 10×: tenfold recommended dosage.

## Data Availability

Data available in a publicly accessible repository The data presented in this study are openly available in [repository name e.g., FigShare] at [doi], reference number [reference number].

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
