# Peer review of "Identification and Dissipation of Chlorpyrifos and Its Main Metabolite 3,5,6-TCP during Wheat Growth with UPLC-QTOF/MS"

_metabolites, 2022, doi:10.3390/metabo12121162_

Round 1
Reviewer 1 Report
The manuscript entitle ‘Identification and dissipation of chlorpyrifos and its main metabolite 3,5,6-TCP during wheat growth by UPLC-QTOF/MS’ presents data of chloropyrifos metabolite. They analyzed and identified chlorpyrifos metabolites in different parts of wheat in full-scan MSE mode and established a chlorpyrifos metabolite screening library using UNIFI software. The initial residues of chlorpyrifos in different parts of the wheat were different, which was in the order leaf > wheat ear > stem. The degradation rate of chlorpyrifos under field conditions is relatively fast, and the half-life value is 2.33-5.05 days.
Data is presented mainly as methodology manuscript. Discussion needs to be strengthened to have valuable outcomes. Few points are added here:
· The concentration of 3,5,6-TCP is affected by multiple factors such as its own properties, natural conditions and crop characteristics and weather conditions.
· In current study, two years 2017-2018. But correlation of 3,5,6-TCP with year and locality in context of weather conditions in both years and during crop growth is missing. Add this data to further strengthen the manuscript.
· Similarly, in connection to above comment weather, locality and crop growth should be correlated with decay of chlorfyrifos and 3,5,6-TCP.
· L 21, Page 9: Specify the locations, not mentioned in methodology. Their weather conditions if are different, and accordingly correlation with degradation and concentration of 3,5,6-TCP.
· Chlorpyrifos can be degraded in 20 metabolites. CPO can be another putative unwanted metabolite that can stay longer than both chlorpyrifos and TCP. Why this was ignored?
· Data presented in Figure 5 is already published fact. It can be presented in form of formula if necessary to present along with reference.
· Figure 2: At 5x in 2018, decay is irregular, what could be the reason?
· Introduction must be improved and provide data on related methodology to clarify what is the main objectives the current study. Discussion and Conclusion should be elaborated with key results.
Author Response
1. The concentration of 3,5,6-TCP is affected by multiple factors such as its own properties, natural conditions and crop characteristics and weather conditions. In current study, two years 2017-2018. But correlation of 3,5,6-TCP with year and locality in context of weather conditions in both years and during crop growth is missing. Add this data to further strengthen the manuscript. Similarly, in connection to above comment weather, locality and crop growth should be correlated with decay of chlorfyrifos and 3,5,6-TCP. L 21, Page 9: Specify the locations, not mentioned in methodology. Their weather conditions if are different, and accordingly correlation with degradation and concentration of 3,5,6-TCP.
Response: Thanks for the comments. We have added the weather conditions in both years in Line 145-150 of methodology to further strengthen the manuscript. Meanwhile, considering the influence of these external factors (pesticide properties, natural conditions, crop characteristics and weather conditions), we designed a 2-year field experiment in the same dimensional area, in order to reduce the experimental errors brought by environmental factors and have enough repetitions, so as to reduce the influence of external environment on the experimental results.
2. Chlorpyrifos can be degraded in 20 metabolites. CPO can be another putative unwanted metabolite that can stay longer than both chlorpyrifos and TCP. Why this was ignored?
Response: Thanks for the comments. For all the possible metabolites of chlorpyrifos reported in Table 2 (including CPO), we established a metabolite screening library for all suspected metabolites of chlorpyrifos in different plant species to detect and identify the possible metabolites of chlorpyrifos during wheat growth. Then we identified the 3,5,6-TCP in different parts of wheat based on accurate mass, retention time, adducts, diagnostic ions and other standards. We did not find the metabolite CPO during wheat growth.
3. Data presented in Figure 5 is already published fact. It can be presented in form of formula if necessary to present along with reference.
Response: Thanks for the comments. We have added the relevant reference [14, 35] in line 437. We felt that the picture format can more intuitively describe the process of hydrolytic metabolism of chlorpyrifos, so we decided to keep the picture.
4. Figure 2: At 5x in 2018, decay is irregular, what could be the reason?
Response: Thanks for the comments. Pesticides applied in the field can undergo metabolic transformation through oxidation, reduction, hydrolysis, photolysis, biodegradation and other forms in plants through the action of enzymes or the influence of external environmental factors (Line 393-396). Influenced by a variety of factors, the decay of 5-fold the recommended dose in 2018 is irregular.
5. Introduction must be improved and provide data on related methodology to clarify what is the main objectives the current study. Discussion and Conclusion should be elaborated with key results.
Response: Thanks for the comments. Data on related methodology and the main objective of the current study have been provided in Line 91-109 and Line 117-123. In addition, we added the key results in Line 446-452.

Reviewer 2 Report
In this study, the authors conducted a study of the degradation and metabolism of chlorpyrifos during wheat growth by spraying plants with different doses of chlorpyrifos. The identification and determination of chlorpyrifos and its main metabolite 3,5,6-TCP during wheat growth was performed by UPLC-QTOF/MS. Degradation kinetics was also studied. The paper seems to be quite interesting. The discussion was rather well described and presented. However, I have some remarks and recommendations:
1.Information on linearity (the investigated concentration range) and selectivity assessments should be added to the description of the validation method.
2.What method was used for calibration?(external or internal standard method?)
3.Were weather conditions (such as temperature and precipitation) taken into account during the study? Were parameters such as temperature adequately monitored? The weather conditions affect the degradation rate of pesticides in plants. For example, the air temperature has an effect on the intensity of metabolic processes in the plant, which determine the degradation rate of pesticides.
4.The tables in the text should be renumbered. The authors started with Table 2 but they should start with Table 1.
5.There are some typos in the text. The text of the manuscript should be carefully checked.
Author Response
1. Information on linearity (the investigated concentration range) and selectivity assessments should be added to the description of the validation method.
Response: Thanks for the comments. We had added the information on linearity and selectivity assessments in Line 216-217 and 271-273.
2. What method was used for calibration?(external or internal standard method?)
Response: Thanks for the comments. External standard method was used for calibration in this study.
3. Were weather conditions (such as temperature and precipitation) taken into account during the study? Were parameters such as temperature adequately monitored? The weather conditions affect the degradation rate of pesticides in plants. For example, the air temperature has an effect on the intensity of metabolic processes in the plant, which determine the degradation rate of pesticides.
Response: Thanks for the comments. We indicated that pesticides applied in the field can undergo metabolic transformation through oxidation, reduction, hydrolysis, photolysis, biodegradation and other forms in plants through the action of enzymes or the influence of external environmental factors (temperature, precipitation, humidity, etc), to produce specific metabolites in Line 393-396. Considering the influence of these external environmental factors, we designed a 2-year field experiment, in order to reduce the experimental errors brought by environmental factors and have enough repetitions, so as to reduce the influence of external environment on the experimental results.
4. The tables in the text should be renumbered. The authors started with Table 2 but they should start with Table 1.
Response: Thanks for the comments. Modified.
5. There are some typos in the text. The text of the manuscript should be carefully checked.
Response: Thanks for the comments. We have checked the manuscript carefully and modified.

Round 2
Reviewer 1 Report
Authors have incorporated the changes appropriately. Still minor spell check and grammar is needed.
Author Response
Point-by-point response to the reviewers’ comments
Reviewer #1:
- Authors have incorporated the changes appropriately. Still minor spell check and grammar is needed.
Response: Thanks for the comments. We have checked the spell and grammar of the manuscript carefully and modified.
Line 25: study→investigate
Line 31: showed a→exhibited a
Line 38-39: showed a trend of increasing first and then decreasing over time; it reached→showed an increasing trend first and then decreasing over time. It reached
Line 83: of CPO is longer than→of CPO was longer than
Line 84: in plants are much higher→in plants were much higher
Line 101: to identify and quantify 50 pesticides in fruits, which has achieved high sensitivity→to identify and quantify 50 pesticides in fruits with high sensitivity and accuracy
Line 201: Spectral data were acquired in a→Spectral data was acquired in a
Line 237: The retention time of chlorpyrifos is 11.20 min→The retention time of chlorpyrifos was 11.20 min
Line 240: 3,5,6-TCP is 6.00 min→3,5,6-TCP was 6.00 min
Line 287: It can be seen from Table 4 that in the field→As presented in Table 4, in the field
Line 305: wheat plant also showed a trend of gradual degradation over time→wheat plant also showed a gradual degradation trend over time
Line 319: being unaffected by changes→irrelevant to changes
Line 321: of wheat is relatively→of wheat was relatively
Line 323: stems are 2.51–4.44→stems were 2.51–4.44
Line 324: degradation is fastest→degradation was fastest
Line 359: parts of wheat are→parts of wheat were
Line 361: groups is basically→groups was basically
Line 375: ; however, the stem→. However, the stem
Line 411: but 3,5,6-TCP→while 3,5,6-TCP
Line 426-429: Racke (1993) used isotope-labeled 36Cl-chlorpyrifos and found that chlorpyrifos is mainly metabolized into 3,5,6-TCP in the soil, and then used 14C markers for further verification and found that 3,5,6-TCP is mineralized and forms carbon oxides as secondary metabolites [34].→Racke (1993) found that chlorpyrifos was mainly metabolized into 3,5,6-TCP in the soil using isotope-labeled 36Cl-chlorpyrifos, and further verified that 3,5,6-TCP was mineralized and forms carbon oxides as secondary metabolites using 14C markers [34].
Line 429: chlorpyrifos will undergo→chlorpyrifos underwent hydrolysis
Line 439: ; as→. As
Line 450: then decreasing over time→then a decreasing trend over time
